# A Lightweight and Efficient Multi-Type Defect Detection Method for Transmission Lines Based on DCP-YOLOv8

**DOI:** 10.3390/s24144491

**Published:** 2024-07-11

**Authors:** Yong Wang, Linghao Zhang, Xingzhong Xiong, Junwei Kuang, Siyu Xiang

**Affiliations:** 1School of Automation and Information Engineering, Sichuan University of Science and Engineering, Yibin 644000, China; 322085404206@stu.suse.edu.cn; 2Research Institute of Electric Power Science, State Grid Corporation of Sichuan Province, Chengdu 610095, China; zhanglh9317@sc.sgcc.com.cn (L.Z.);

**Keywords:** transmission lines, defect detection, machine vision, YOLOv8, lightweight, deformable convolution

## Abstract

Currently, the intelligent defect detection of massive grid transmission line inspection pictures using AI image recognition technology is an efficient and popular method. Usually, there are two technical routes for the construction of defect detection algorithm models: one is to use a lightweight network, which improves the efficiency, but it can generally only target a few types of defects and may reduce the detection accuracy; the other is to use a complex network model, which improves the accuracy, and can identify multiple types of defects at the same time, but it has a large computational volume and low efficiency. To maintain the model’s high detection accuracy as well as its lightweight structure, this paper proposes a lightweight and efficient multi type defect detection method for transmission lines based on DCP-YOLOv8. The method employs deformable convolution (C2f_DCNv3) to enhance the defect feature extraction capability, and designs a re-parameterized cross phase feature fusion structure (RCSP) to optimize and fuse high-level semantic features with low level spatial features, thus improving the capability of the model to recognize defects at different scales while significantly reducing the model parameters; additionally, it combines the dynamic detection head and deformable convolutional v3’s detection head (DCNv3-Dyhead) to enhance the feature expression capability and the utilization of contextual information to further improve the detection accuracy. Experimental results show that on a dataset containing 20 real transmission line defects, the method increases the average accuracy (mAP@0.5) to 72.2%, an increase of 4.3%, compared with the lightest baseline YOLOv8n model; the number of model parameters is only 2.8 M, a reduction of 9.15%, and the number of processed frames per second (FPS) reaches 103, which meets the real time detection demand. In the scenario of multi type defect detection, it effectively balances detection accuracy and performance with quantitative generalizability.

## 1. Introduction

With rapid economic development and increasing energy demand, transmission lines play a vital role in the power system. However, defects and failures in transmission line equipment may lead to power outages, causing significant impacts on the economy and daily life. Therefore, it is crucial to regularly inspect transmission line equipment to detect and repair defects promptly. In recent years, deep learning-based object detection algorithms have been widely used in transmission line defect detection and have achieved remarkable results. However, deploying these algorithms in resource-limited edge environments remains challenging because they usually have high computational complexity and many model parameters. To address this problem, researchers have begun to focus on the study of lightweight deep learning algorithms, aiming to reduce the number of model parameters and computational complexity while maintaining high detection accuracy to meet the needs of real-time detection. However, lightweight models may have difficulties in identifying a small number of defects. With limited computing resources, the structure and number of parameters of the model are greatly reduced, resulting in the model lacking sufficient representation ability to capture the subtle visual differences between different defects. For example, defects in transmission lines may include cracks, corrosion, hanging foreign objects, etc. Each defect requires the model to be able to recognize specific shapes, textures, and contextual information. The lightweight model is limited in complex feature extraction and abstraction levels, which affects its performance in identifying multiple defect types.

Object detection algorithms are mainly divided into two detection methods: single-stage detection and two-stage detection. The two-stage network R-CNN usually provides higher detection accuracy because the two-stage method can adjust the bounding box position more finely. However, due to the two-stage processing process, R-CNN is usually more complex and computationally more expensive than the single-stage algorithm. For example: In [1], the authors proposed a Faster R-CNN model with a feature enhancement mechanism for corrosion detection in transmission line fitting. The average accuracy of rust detection is 97.07%, but the calculation is very complicated. The work in [2] introduced an improved multi-target defect recognition algorithm based on Faster R-CNN. The average accuracy of this model is 86.57%, and the average detection time is 0.307 s, but this method is only applicable to a few defect detections. Researchers are still committed to developing lightweight R-CNN models. For example, the authors in [3] reduced the model size from 245 MB to 47.1 MB by adjusting the backbone and head structure of Mask R-CNN, but it is still very large. The work in [4] used MobileNet to build the Faster R-CNN model backbone network, which effectively reduces computing costs and the detection time of each picture is only 0.05 s. Although the above-mentioned Faster R-CNN target detection algorithms have made significant progress in improving detection accuracy, these advances usually come at the expense of increased model parameters and increased computational costs. Even if these algorithms are optimized for lightweight structures, they still face challenges in achieving real-time detection.

Compared with the two-stage detection algorithm, the single-stage detection algorithm is more suitable for scenarios that require fast detection. The authors in [5] used the MobileNet lightweight network to optimize YOLOv4, and used transfer learning to train and verify the improved model, but the model still requires high computing resources. The study in [6] proposed an ultra-lightweight and ultra-fast target recognition network based on adaptive feature fusion. The model size is only 3.6 MB, but the detection accuracy needs to be improved. The authors in [7] proposed a new lightweight model YOLOv7-Tiny, which effectively improves the recognition ability of small targets, but only for the detection of defects in a few categories. The DPNet proposed in [8] combines the lightweight characteristics of a single path and the advantages of a dual path. It can extract high-level semantic features and low-level object details in parallel, thereby improving detection accuracy, but its cross-domain generalization remains to be considered.

Previous studies have made improvements to specific types of grid equipment defect datasets [9,10], but their generalization capabilities are limited. Since different types of defects have unique characteristics, with significant differences in size, color, structure, shape, etc., the research results using a single type of dataset are difficult to apply to diverse scenarios. To improve the generalization of the algorithm, researchers need to develop lightweight algorithms that can adapt to a variety of defect types, which requires the designed model structure and feature extraction method to be more robust and versatile.

Therefore, while studying lightweight algorithms in the field of power grid equipment inspection, maintaining a high detection accuracy for multiple types of defects has become a technical problem that urgently needs to be solved. Researchers need to find a balance between the lightweight structure and the detection accuracy to ensure the efficiency and accuracy of real-time detection. At the same time, developing new model structures and training strategies to enhance the detection capabilities of multiple types of defects while maintaining the lightweight of the model is the key to achieving this goal.

To address these challenges, we propose a lightweight and efficient multi-type defect detection method for transmission lines based on DCP-YOLOv8, and our main research contributions are summarized as follows:To improve the model’s ability to extract features of different types of defects, an advanced semantic information extraction module with strong adaptive ability, DCNv3, is introduced. This module is used to replace the C2f module in the backbone part of the YOLOv8 algorithm. It can effectively improve the model’s ability to detect defects. It also helps the model to accurately recognize defects in complex scenes.A re-parameterized cross-stage feature fusion Partial Network (RCSP) is designed, which implements feature re-parameterization technology in each cross-stage feature fusion network stage. Through its loop design and unique combination of shared weights and independent weight convolutional layers, multi-scale features are effectively fused to improve the model’s performance in feature extraction, context understanding, and detail preservation. This structure not only significantly reduces the number of parameters of the model, but also maintains high accuracy, robustness, and generalization in defect detection tasks, achieving lightweight goals.A DCNv3-Dynamic Head inspection head combining a dynamic inspection head and a deformable convolutional v3 is developed. In real-life scenarios, defects in transmission lines exhibit diversity and variability, including in the type, shape, and size of defects. To enhance the model’s adaptability and generalization ability to these different variations, the DCNv3-DynamicHead can effectively improve the model’s performance in detecting defects of multiple types of transmission lines, and thus effectively cope with the challenge of detecting defects that are complex and variable.

To validate the effectiveness of the method described in this paper, compared to most of the current transmission line defect detection methods that only detect a few defect categories, we trained and evaluated the model on a dataset that covers seven major categories of common ancillary facilities of transmission lines (including large-size fixtures, small-size fixtures, appurtenances, poles, towers, foundations, insulators, and passages), which contains seven major categories and 20 subcategories of different defects, thus ensuring the diversity and comprehensiveness of the model.

The rest of the paper is organized as follows. Section 2 describes the proposed method in detail, including the network architecture and improvement strategies. Section 3 describes the experimental setup, including the experimental environment, experimental data, and evaluation metrics. Section 4 presents the experimental results and analysis, Section 5 shows some limitations of the proposed methodology in this paper, and finally, Section 6 summarizes the whole paper.

## 2. Model Construction Based on DCP-YOLOv8

Deep learning target detection algorithms have developed rapidly in recent years. The R-CNN series [11,12,13], MaskR-CNN [14], and YOLO series [15,16,17,18] are the two most influential methods. Faster R-CNN improves detection efficiency by introducing the Region Proposal Network (RPN) and combining it with the Feature Pyramid Network (FPN) to improve multi-scale detection performance. In addition, SSD [19] and RetinaNet [20] also show excellent performance. As a single-stage algorithm, YOLO is dedicated to the real-time detection and prediction of target categories and locations directly in images. The YOLO series has been iterated in multiple versions, such as YOLOv3, YOLOv4, YOLOv5, and YOLOv8, to continuously optimize the model structure and feature extraction, and enhance the detection speed and accuracy. For example, YOLOv4 uses the CSPDarknet53 backbone network and Mosaic data enhancement, while YOLOv5 further improves the lightweight capability of the model. These advances have made target detection algorithms widely used in many fields. YOLOv8 is the latest version of the YOLO series, launched in 2023. It is optimized based on YOLOv5. Both adopt CSP and SPPF modules, but YOLOv8 introduces the C2f structure to enhance the gradient flow and adjusts the number of channels to adapt to different scale models. The detection head of YOLOv8 changed from coupled to decoupled, separated classification and detection, and changed from Anchor-Based to Anchor-Free. Its core structure consists of three parts: Backbone, Neck, and Head: The Backbone is responsible for feature extraction, the Neck fuses different levels of features through the PANet structure, and the Head performs final classification and bounding box regression to improve detection accuracy and speed.

The baseline YOLOv8n model has some deficiencies in feature extraction, which is mainly due to the following reasons: (1) As a lightweight model, the design of YOLOv8n emphasizes reducing computing resources and improving speed, which usually means that the model has lower complexity and depth. This simplification may hinder the model’s ability to extract abstract features at a higher level, especially when subtle signals of different types of defects need to be distinguished. (2) To reduce the computational burden, lightweight models tend to quickly reduce the resolution of feature maps during feature extraction. This downsampling may lose key spatial details and affect the effect of subsequent feature fusion. (3) To remain lightweight, YOLOv8n has a relatively small number of parameters, which means that the model’s learning capacity is limited and it may not be able to fully learn the complex distribution of multiple types of defects.

To maintain high defect detection accuracy while reducing the number of model parameters, we improved the YOLOv8 algorithm and generated the DCP-YOLOv8 model. First, the convolution in the C2f module of the YOLOv8 backbone network is replaced with deformable convolution (DCNv3), thereby enhancing the algorithm’s generalization ability to detect transmission line defects of different scales. Secondly, we propose a re-parameterized cross-stage feature fusion structure (RCSP) to optimize the neck part of the model. This re-parameterization helps to better integrate features at different scales. Finally, we design a DCNv3-Dyhead detection head. By integrating the dynamic detection head (DynamicHead) and deformable convolution (DCNv3), the model has better adaptability and generalization capabilities and can handle various complexities. defect detection tasks. The proposed model structure is shown in Figure 1.

### 2.1. C2f_DCNv3 Module

The DCNv3 [21] used in this article is improved based on DCNv2 [22]. Compared with DCNv2, DCNv3 borrows the idea of depth-separable convolution [23] to achieve weight sharing between convolutional neurons. By introducing a multi-group mechanism, different groups on a single convolutional layer can have different spatial aggregation patterns, thereby bringing stronger features to downstream tasks. In addition, DCNv3 also adopts the method of normalizing the modulation scalar along the sampling point, which alleviates the gradient instability problem in the DCNv2 layer when using large-scale parameters and data for training. The complete DCNv3 calculation formula is as follows:(1)$$y\left(p_{0}\right)=\overset{G}{\underset{g=1}{\sum }}\overset{K}{\underset{k=1}{\sum }}w_{g}m_{gk}x_{g}\left(p_{0}+p_{k}+\Delta p_{gk}\right)$$

In the formula, G represents the number of groups. $w_{g}$ represents the projection weight shared within each group, and $m_{gk}$ represents the normalized modulation factor of the k-th sampling point in the g-th group, which is normalized along the dimension k by the softmax function. $p_{0}$ represents each position on the output feature map y, $p_{k}$ represents the predefined sampling point, and $△p_{gk}$ represents the offset of each sampling point. The purpose of using the DCNv3 module is to enhance the model’s ability to extract different features. DCNv3 achieves this by learning the offset of each sample point of the convolution kernel, allowing it to adapt to the geometry of the object. DCNv3 enhances the feature extraction capabilities of transmission line defect targets of different scales by learning different optimal convolution kernel structures for different target data. The structure of Deformable Convolutions v3 is shown in Figure 2.

We integrated the DCNv3 operator into the C2f structure to form the C2f-DCNv3 structure, as shown in Figure 3. We noticed that the DCNv3 operator not only improves the limitations of traditional convolution in handling long-term dependencies and spatial adaptive aggregation, but also, the characteristics based on sparse sampling are more suitable for defect detection tasks at different scales. Due to sparse sampling, DCNv3 only requires a 3 × 3 kernel to learn long-range dependencies, which is easier to optimize. This operator inherits the inductive bias of traditional convolutions (CNN), and we can achieve a better trade-off between computational complexity and accuracy. Through experiments, it was found that the addition of the DCNv3 operator improved network performance, but too many DCN layers would cause performance to decrease instead of increase, slow down the speed, and increase the difficulty of parameter adjustment. Replacing the C2f module of the backbone network with the C2f_DCNv3 module can achieve the best performance. This change enables the network to adaptively adjust the receptive field and more accurately capture the shape and size of objects, and it enhances the robustness of the network.

### 2.2. Re-Parameterized Cross-Stage Feature Fusion Partial Network (RCSP)

In deep learning, to improve the accuracy and robustness of target detection, it is critical to ensure that the detector can fully understand and integrate different levels of feature information. This feature information includes both high-level semantic recognition of objects and low-level perception of the spatial location of objects, which makes the detector neck an important part of the entire framework [24,25]. The traditional FPN network [26] is effective in multi-scale feature fusion, but it may lose detailed information, take a long time to train, and have a strong dependence on context information. To solve these problems, we designed the RCSP structure, which is based on the design of CSPNet [27]. CSPNet maximizes the difference in gradient combinations through some transition layers and adopts a strategy of truncating the gradient flow to prevent the reuse of gradient information. However, designing an appropriate degree of fusion of transition layers is a challenge. Excessive fusion may lead to information loss, while insufficient fusion may not maximize the gradient difference.

The RCSP structure we designed is improved based on the CSPNet structure and is designed to cope with the defect detection task in the complex background of transmission lines. The structural process is as follows: first, the input is passed through two CBS modules to generate two feature maps, and then the feature map y2 is passed through a loop list containing multiple convolutional layers. In the loop, for even-indexed convolutional layers, shared weights are used to process y2; for odd-indexed convolutional layers, independent weights are used to process y2. Each iteration adds the currently processed feature map y1 to the list, each feature map y1 retains its original feature information, and finally, these different levels of feature maps are spliced together along the channel dimension. This operation does not introduce new parameters and can effectively reduce the number of parameters of the model. The RCSP structure flow chart is shown in Figure 4.

The RCSP structure adopts a loop processing mechanism to retain feature information at all levels from the original image to the gradually abstracted feature map by iteratively refining the feature map. Connecting these feature maps in the channel dimension allows the model to learn and extract features at different levels while obtaining local details and global context information, which helps the model understand the scene more comprehensively. In addition, the use of circular lists helps retain and transmit original feature information, reduces information loss caused by convolution operations, and improves the model’s defect detection capabilities.

The RCSP structure provides the model with a feature representation that is both general and refined through its unique combination of shared weights and independently weighted convolutional layers. This structure enables the model to further optimize specific features through independent weighted convolutional layers based on a shared feature space. This design strategy allows the model to extract features at different levels, with the complementary characteristics of shared weights and independent weights, allowing the model to reduce parameters while improving computational efficiency, thereby improving its performance.

### 2.3. DCNv3-Dyhead Module

We adopt a dynamic head module [28] to unify the attention mechanism. As shown in Figure 5, dynamic head blocks composed of scale awareness, space awareness, and task awareness are stacked in sequence. The advantage of dynamic detection heads in transmission line defect detection is their ability to:Adapt to defects of different scales: By adjusting the scale of the feature map, the moddl can capture defects of various sizes from small to large.Locating defects in the image: Spatial awareness enables the model to focus on any location where defects appear in the image.Process task-specific information: Task-aware capabilities help the model better identify and process key information such as the type, location, and shape of defects.

This structure not only improves the accuracy of defect detection, but also does not add additional computational burden, so it is efficient and practical in practical applications. We fuse DCNv3 with a dynamic detection head, combining the dynamic shape adjustment of deformable convolution and the scale perception of the dynamic detection head to more effectively identify defects of different sizes; in addition, depth-separable convolution provides powerful feature expression in deformable convolution V3, working together with the dynamic detection head to effectively improve recognition accuracy. The fusion structure DCNv3-Dyhead also enhances the model’s ability to identify unknown defects, which is beneficial to its generalization in practical applications.

**Figure 5 sensors-24-04491-f005:**
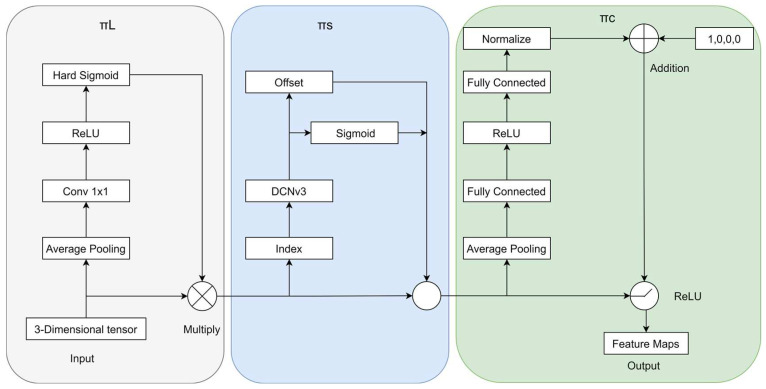
Structure of the DCNv3-Dyhead.

### 2.4. Loss Function

We use the same loss function as the original YOLOv8, which consists of two parts: category classification loss and bounding box regression loss. The category classification loss uses binary cross entropy (BCE) loss. Compared with VFL loss, BCE loss is easier to implement and has an equivalent effect. The bounding box regression loss uses distributed focus loss (DFL) [29] and CIoU [30]. The formula is as follows:(2)$$L_{yolov8}=L_{cls}+L_{reg}=L_{BCE}+L_{DFL}+L_{CIoU}$$ where $L_{cls}$ represents the category classification loss and $L_{reg}$ represents the bounding box regression loss. YOLOv8 uses BCE (Binary Cross Entropy) as the classification loss. Each category determines whether it is of this type and outputs the confidence level. The formula is as follows:(3)$$L_{BCE}=-w\left[y_{i}\log\left(x_{i}\right)+\left(1-y_{i}\right)\log\left(1-x_{i}\right)\right]$$
where w is the weight, $y_{i}$ is the label value, and $x_{i}$ is the predicted value of the model. DFL is an optimized version of the focal loss function. The formula is as follows:(4)$$L_{DFL\left(S_{i},S_{i+1}\right)}=-(\left(y_{i+1}-y\right)\log\left(S_{i}\right)+\left(y-y_{i}\right)\log\left(S_{i+1}\right))$$

The meaning of DFL is to optimize the probability of the two positions closest to label y in the form of cross-entropy, that is, the distribution obtained by network learning is theoretically near the real floating point coordinates, and in linear interpolation mode, we obtain the weight of the distance to the left and right integer coordinates, allowing the network to focus on the distribution of nearby areas of the target location faster. CIOU incorporates the aspect ratio based on DIOU. CIOU loss includes overlapping area, center point distance, and aspect ratio. The formula is as follows:(5)$$L_{CIoU}=1-IOU\left(A,B\right)+\frac{\rho^{2}\left(A_{ctr},B_{ctr}\right)}{c^{2}}+av$$
(6)$$v=\frac{4}{\pi^{2}}{\left(\tan^{-1}\frac{w^{gt}}{h^{gt}}-\tan^{-1}\frac{w}{h}\right)}^{2}$$
(7)$$a=\frac{v}{\left(1-IOU\right)+v}$$
where a is the weight function, v is used to measure the consistency of the aspect ratio, A and B are the prediction box and the basic box, respectively, $A_{ctr}$ and $B_{ctr}$ are the center points of A and B, respectively, ρ(·) is the Euclidean distance, and c is the diagonal length of the minimum bounding box that covers two boxes.

## 3. Experiment Setup

### 3.1. Experiment Environment

In the experiment, we used Python 3.9 as the programming language and Pycharm as the integrated development environment. The hardware parameters used in the experiment are shown in Table 1.

### 3.2. Datasets and Evaluation Metrics

The dataset used in this paper is derived from real inspection images of UAV aerial photography of several transmission lines of State Grid Sichuan Electric Power Company (Chengdu, China). The annotations of the dataset are all reviewed by technical experts in the initial bidding and business experts, and the samples are of high quality. The dataset covers seven categories of common supporting facilities of transmission lines: large-size fixtures, small-size fixtures, ancillary facilities, towers, foundations, insulators, and passages, and contains 20 specific types of defects, such as vibration-proof hammer shifting, insulator self-detonation, deformation of small metal fittings, blockage of the passageway, abnormal bolts in joints, broken vibration-proof hammer, corrosion of vibration-proof hammer, broken equalizing ring, bird’s nest of pole tower, dislodging of the equalizing ring, damage of appurtenances, hornet’s nest of pole tower, debris in passageway, insulator breakage, deformation of vibration-proof hammer, damage of infrastructures, deformation of overhanging wire clamps, corrosion of overhanging wire clamps, corrosion of small metal fittings, and filthiness of insulators.

Within these broad categories, defects under the same category usually have a high degree of similarity. For example, in the category of large-size gold fixtures, this may include a variety of corrosion or abrasion problems caused by long-term exposure to natural environments. These problems may visually appear as similar texture changes or color degradation, allowing the model to learn common features that distinguish these defects during training. On the other hand, defects between different broad categories show greater variability. For example, defects such as insulator self-explosion usually show surface cracks or breaks, while bird nests and channels involve the recognition of the external environment of the transmission line. These defects have large differences in the representation and contextual information in the images, which helps the model to learn the key features to distinguish between different defect types.

By including different defects of these seven major categories and 20 subcategories, the dataset not only ensures the diversity and comprehensiveness of the model but also helps to improve the generalization ability and robustness of the model in real scenarios. Such a dataset provides powerful data support for UAV inspection of transmission lines, which is of great significance for improving the safe operation and maintenance efficiency of transmission lines. The distribution of the number of defective samples is shown in Figure 6. The distribution of the number of samples varies greatly, which is closer to the real grid production environment conditions. We divided the dataset into a training set (6388 images), a validation set (1204 images), and a testing set (415 images) according to 8:1.5:0.5. In the case of limited data resources, the main reasons for dividing the data set in this way are as follows. 1. Due to the uneven distribution of data set samples, to ensure that the training set can contain all samples, we choose to set the training set ratio to 80%. 2. The size of the validation set should usually be large enough to reflect the model’s performance on real data. Setting its ratio to 15% can provide enough samples for effective hyperparameter adjustment and model verification, ensuring the trained model has good generalization ability. 3. The main purpose of the test set is to evaluate the model’s performance on unseen data. Usually, a large sample size is not required, as long as it can cover samples of all categories. Setting the test set ratio to 5% is sufficient.

We set some key parameters in the model training process according to the experimental equipment conditions and model training experience, as shown in Table 2. In order to accelerate model convergence, mosaic data enhancement was turned off in the last 10 epochs of the training process.

We use stochastic gradient descent (SGD) as the optimizer with an initial learning rate of 0.01 and a momentum of 0.9. We use mAP50, F1 score, and FPS as evaluation indicators of the model’s detection performance. We choose the number of parameters and FLOPs to evaluate the complexity and size of the model. The specific calculation formula is as follows:(8)$$P=\frac{TP}{TP+FP}$$
(9)$$R=\frac{TP}{TP+FN}$$
(10)$$F1=\frac{2\cdot P\cdot R}{P+R}$$
(11)$$AP=\overset{1}{\underset{0}{\int }}PdR$$
(12)$$mAP=\frac{\sum_{i=1}^{N}AP_{i}}{N}$$
(13)$$FPS=\frac{1}{inference\;time\;per\;frame}$$

TP is the number of positive samples that are correctly identified as positive samples; FP is the number of negative samples that are incorrectly identified as positive samples; FN is the number of positive samples that are incorrectly identified as negative samples; N is the total number of detection target categories; AP is the under the P-R curve area; and mAP is the average of the total number of APs of various defects detected. FPS represents the number of image frames processed in one second. The inference time per frame is the time required to process a single frame using the YOLOv8 model.

## 4. Experiment Results and Analysis

### 4.1. Comparison of the Effects of C2f_DCNv3 Module at Different Positions in the YOLOv8 Model

In the YOLOv8 model, as the network layer deepens, the size of the feature map will gradually decrease. Since the size of the feature map at different locations is different, this may affect the performance of the C2f_DCNv3 structure at different locations. To determine the best placement of the C2f_DCNv3 structure, the researchers conducted comparative experiments in four models: the original YOLOv8n model, the model used only in the backbone, the model used only in the neck, and the model used in both the backbone and the neck. The experimental results are shown in Table 3.

The experimental data in Table 3 show that stacking too many DCNv3 layers does not always lead to performance improvement. Using the C2f_DCNv3 structure in both the backbone and the neck leads to an increase in the number of model parameters and a decrease in detection accuracy instead of an increase. However, using the C2f_DCNv3 structure in the backbone alone or the neck alone resulted in improved model performance. Among them, the use in the backbone part alone has the best effect, as shown in the following: the mAP@0.5 is increased by 1.5%, the number of parameters is increased by 0.2 M, and the computational amount is reduced by 0.5 G. Therefore, the C2f_DCNv3 module effectively enhances the detection capability of the YOLOv8n algorithm for transmission line defects.

### 4.2. Comparison Experiments with Different Feature Fusion Modules

To verify the lightweight effect of the RCSP module, this section conducts comparative experiments using the original YOLOv8n’s PANet module, the CSPNet module, and the RCSP module, respectively. The experimental results are shown in Table 4.

As can be seen from Table 4, the RCSP model performed well in this comparison experiment. The concept of heavy parameterization allowed the model to obtain local details and global context information at the same time and significantly reduced the number of model parameters, and the mAP@0.5 increased. 1.2%. In addition, compared with the CSPNet module, the RCSP module significantly reduces the number of parameters by 1.1 M, and the average accuracy is also improved. The lightweight effect is very well achieved. These results show that the RCSP module is an effective feature fusion strategy and improves the efficiency of the model.

### 4.3. Comparison Experiments of Different Detection Head Modules

We aim to verify whether the design of the DCNv3_Dyhead detection head module can improve the model’s performance in detecting multiple types of defects on transmission lines, and thus, effectively meet the challenges of detecting complex and changing defects. In this section, comparison experiments are conducted using the original YOLOv8 detection head module, the dynamic detection head Dyhead module, and the DCNv3_Dyhead module, respectively. The experimental results are shown in Table 5.

The experimental results in Table 5 show that the detection performance of the YOLOv8 model can be improved by improving the detection head. Compared with the dynamic detection head, the fused DCNv3_Dyhead detection head further improves the defect detection accuracy while reducing the number of parameters. Compared with the original YOLOv8n detection head, the mAP@0.5 value has increased by 2.9%, and the number of parameters has also increased. In addition, the model’s F1 score is improved by 3.3% compared to the baseline model. It is proved that the DCNv3_Dyhead detection head is effective in improving the detection performance of the model.

### 4.4. Ablation Experiments

In model improvement experiments, do some new components work? How does the model’s performance change when a component is removed or replaced? To answer these questions, ablation experiments were designed. By systematically adding or removing parts of the model, you can evaluate the contribution of each part to the final performance of the model, and thereby gain a deeper understanding of the working mechanism of the model and enhance the interpretability of the model. All ablation experiments in this section are conducted on the same data set and the same experimental environment, and all convolution training is started from scratch without using pre-trained weight files. The ablation experiment was performed based on YOLOv8n. √ indicates the use of this improved strategy. The experimental results are shown in Table 6.

The experimental results in Table 6 show that by introducing DCNv3 to the backbone network to replace the C2f module, the model feature extraction capability is enhanced, mAP50 is increased by 1.5%, and the parameter amount is increased by 0.2 M. The efficient RCSP module reduces the calculation and memory burden, reduces the number of parameters by 0.7 M, and increases mAP@0.5 by 1.2%, effectively optimizing the algorithm. The DCNv3_Dyhead detection head significantly improves the defect detection accuracy, mAP@0.5 increases by 2.9%, and the number of parameters increases by 2 M. The reason is that the model needs to learn how to identify and locate targets at different scales. Combining C2f_DCNv3 and RCSP, mAP@0.5 is increased by 1.4% and the number of parameters is reduced. Combining C2f_DCNv3 and DCNv3_Dyhead, mAP@0.5 increases by 2.3%, and the number of parameters increases, but mAP@0.5 is less than using DCNv3_Dyhead alone. Combining RCSP and DCNv3_Dyhead, mAP@0.5 is slightly increased, by 1%, and the number of parameters is slightly reduced. The three improved methods are combined to achieve the best detection accuracy. mAP@0.5 is increased by 4.3%, the number of parameters is reduced by 0.3 M, and the calculation amount is increased by 0.8GFLOPs. In terms of FPS, different module combinations also have an impact on the running speed of the model. When only the C2f_DCNv3 module is introduced, FPS drops by 11.5; when only the RCSP module is introduced, FPS increases by 19.1; when only the DCNv3_Dyhead module is introduced, FPS drops by 25.3. When the three improvements were combined, FPS dropped by 6.2. This shows that although the combination of these three modules improves the performance of the model, it also reduces the running speed of the model to a certain extent, but overall, this reduction is acceptable.

In summary, the DCP-YOLOv8 model achieves the goal of improving detection accuracy while maintaining a lightweight effect to some extent. While the addition of some components increases computational complexity and model size, other components help mitigate these effects. The optimal combination of improving accuracy while still maintaining a high FPS shows that the model can achieve a balance of accuracy and lightweight structure without overly sacrificing real-time performance. This trade-off is normal because, in real applications, it is often necessary to find the right balance between accuracy, real-time, and resource consumption.

### 4.5. Comparison Experiments

We aim to verify whether the DCP-YOLOv8 algorithm proposed in this paper can improve the overall accuracy of defect detection for multiple types of defects while maintaining the lightweight effect and realizing the goal of real-time detection. Comparison experiments are conducted between the proposed method and mainstream target detection methods in a dataset containing 20 real transmission line defects, and the experimental results are shown in Table 7.

In comparing the performance of several of the most dominant models in the task of target detection, we consider several aspects such as mean accuracy (mAP50%), frames per second (FPS), floating point operations (FLOPs), and number of parameters.

DCP-YOLOv8 has the highest mAP@0.5 (72.2%) and F1-Score (69.5%), which shows that it is significantly better than other methods in terms of accuracy. The high mAP@0.5 value means that DCP-YOLOv8 performs well in detecting objects of various sizes and shapes. The GFLOPs of DCP-YOLOv8 is 8.9, which is slightly higher than YOLOv8’s 8.1, but still much lower than the values of Faster-RCNN, RetinaNet, and some other methods. This means that DCP-YOLOv8 requires fewer computing resources when performing inference and is more efficient. In terms of parameter number, DCP-YOLOv8 has only 2.8 M, which is the lowest parameter number among all the compared methods, which helps to reduce the memory footprint of the model and facilitates deployment on devices with limited resources. In addition, the FPS of DCP-YOLOv8 is 103.1, which is slightly lower than YOLOv8’s 109.3, but still a very high value, indicating that DCP-YOLOv8 is capable of high-speed real-time inference. This is very important for application scenarios that require fast response, such as video surveillance, autonomous driving, etc.

In summary, the DCP-YOLOv8 offers high-precision detection performance while maintaining a lightweight effect, which gives it a clear advantage in real-world applications. Especially in scenarios where real-time, accuracy, and resource consumption need to be considered simultaneously, DCP-YOLOv8 demonstrates excellent performance, making it a very attractive choice.

In addition, we analyze the changes in box loss, class loss, and mAP@0.5 during training. As shown in Figure 7, we observe that the training loss of DCP-YOLOv8 decreases faster than that of YOLOv8n, and the box loss and class loss are similar. This shows that DCP-YOLOv8 is better than YOLOv8n. The change in mAP also shows that DCP-YOLOv8 performs significantly better than YOLOv8 during training. As can be seen from Figure 6, when training reaches the 140th epoch, the loss curve fluctuates. This is because we turned off Mosaic data enhancement in the last 10 epochs of the training process.

Figure 8 shows the effect of the baseline model and our proposed method on transmission line defect detection on the test set. Compared with the baseline model, the DCP-YOLOv8 model can detect more defective targets, and the detection accuracy has been improved to varying degrees. Thus, the effectiveness of the model in improving the detection accuracy of multiple types of defects in transmission lines is verified.

### 4.6. Robustness Analysis in Complex Environments

In this section, we experimentally verify the robustness of our proposed method in the face of image degradation in complex environments. Images may be degraded due to environmental conditions, the quality of drone camera footage, and image processing technology limitations. Therefore, it is crucial to evaluate the resistance of object detection models to such degradation, which is directly related to their performance in practical application scenarios. Considering that power lines often pass through changing environments, we selected four typical environments for testing: rain, fog, strong light, and dimness. At the same time, to examine the model’s performance under different types of environmental noise, we also evaluated the impact of Gaussian noise.

The detection results of DCP-YOLOv8 in complex environments are shown in Figure 9. It can be seen that the baseline YOLOv8n model performs poorly in foggy, rainy, and dim environments, it cannot identify the bird’s nest covered on the tower, and the detection level drops significantly when facing images containing Gaussian noise. Therefore, the baseline model has low applicability in complex environments and poor robustness to image degradation under complex conditions. Compared with the baseline YOLOv8 model, DCP-YOLOv8 can accurately detect all defective targets in common complex environments, with a higher detection accuracy and better adaptability to these complex environments. In summary, these experiments demonstrate the effectiveness of our proposed method in improving the robustness of the YOLOv8 model in the face of image degradation in complex environments. These results are significant for practical applications where the model must perform accurately under challenging conditions.

### 4.7. Grad-CAM Visual Analysis

In this section, we visually analyze the results of transmission line defect detection using Grad-CAM [35], especially regarding the detection of small objects. In object detection, Grad-CAM can help us understand how the model locates and recognizes the target. By analyzing the Grad-CAM visualizations, we can better understand how much the model focuses on the defect area. The redder the color, the more attention the model pays to that region. We train the YOLOv8n and DCP-YOLOv8 models and evaluate their performance on the test set. For each image in the test set, we generate a Grad-CAM visualization for the bounding box of the predicted transmission line. Figure 8 shows some examples of the baseline model and the Grad-CAM visualization generated by our proposed method. From the observation of the YOLOv8n defective target heat maps in Figure 10a,c,e, we know that the Grad-CAM visualization of the baseline model is less accurate, with less focus on the region and lower confidence. In contrast, the Grad-CAM visualization of our proposed DCP-YOLOv8 method is more focused and more accurately captures the key features of anomalously defective objects. This experiment confirms the advantages of DCP-YOLOv8 in terms of feature extraction, model focus, and localization accuracy, and also demonstrates its good interpretive and generalization capabilities.

## 5. Limitations

Although DCP-YOLOv8 significantly improves the combined accuracy of multi-class defect detection while maintaining fast operational efficiency, this achievement does not come without a price. With the incorporation of the new deformable convolutional module or the DCNv3_Dyhead module, DCP-YOLOv8 has increased in model complexity, which inevitably leads to a decrease in its frames per second (FPS) compared to the original YOLOv8 model. This increased complexity may have an impact on the deployment of the model, especially in environments with limited computational resources. Going forward, we will aim to explore more efficient network components aimed at reducing computational and memory overheads to make the model more lightweight and easier to deploy while maintaining its performance. In addition, our approach is currently validated only on the transmission line dataset, and analyzing its applicability and performance for other target detection tasks requires further empirical studies. We expect to extend the superior performance of DCP-YOLOv8 to a wider range of application scenarios through subsequent work, as well as to open up new paths for model optimization and efficiency improvement.

## 6. Conclusions

A lightweight and efficient multi-type defect detection method for transmission lines based on DCP-YOLOv8 is proposed to identify transmission line defects quickly and accurately. The proposed C2f_DCNv3 module and RCSP module enhance the feature extraction and fusion capabilities, and the DCNv3-Dyhead detection head improves the defect localization accuracy. DCP-YOLOv8 performs well in the transmission line defect detection task, achieving an average accuracy of 72.2% (mAP@0.5), a computational complexity as low as 8.9GFLOPs, a parameter count of only 2.8 M, an F1 score of 69.5%, and a number of processing frames per second (FPS) as high as 103.1, which is ideal for resource-constrained environments. In the future, we will continue to improve the accuracy of the algorithm and promote its application to provide an efficient and reliable solution for transmission line defect detection.

## Figures and Tables

**Figure 1 sensors-24-04491-f001:**
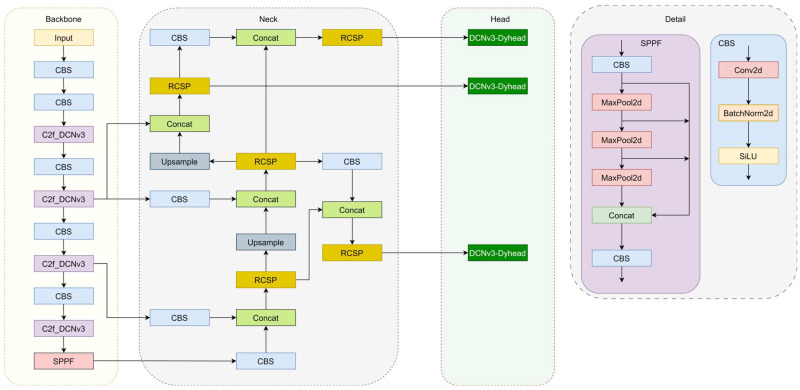
Detailed architecture of the proposed DCP-YOLOv8.

**Figure 2 sensors-24-04491-f002:**
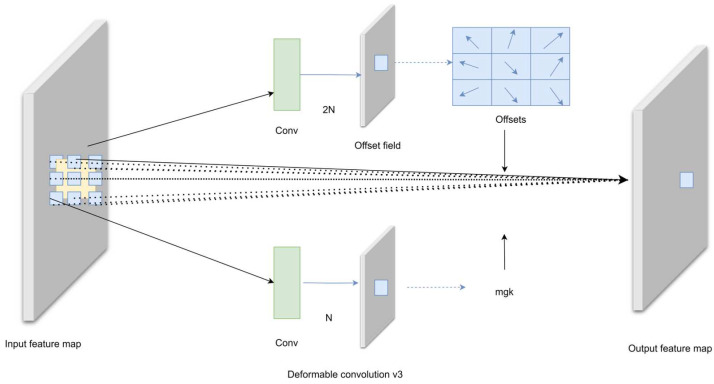
Structure diagram of the Deformable Convolutions v3.

**Figure 3 sensors-24-04491-f003:**
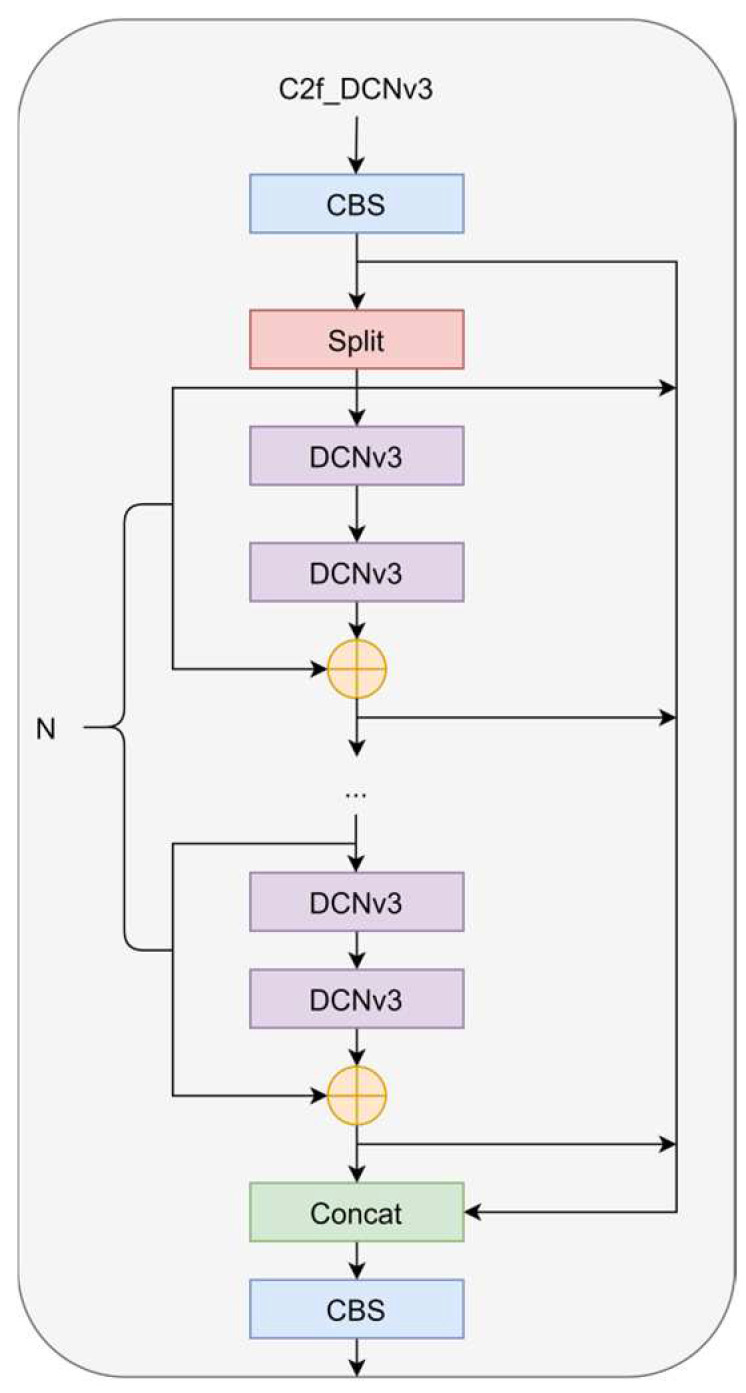
Structure of the C2f_DCNv3.

**Figure 4 sensors-24-04491-f004:**
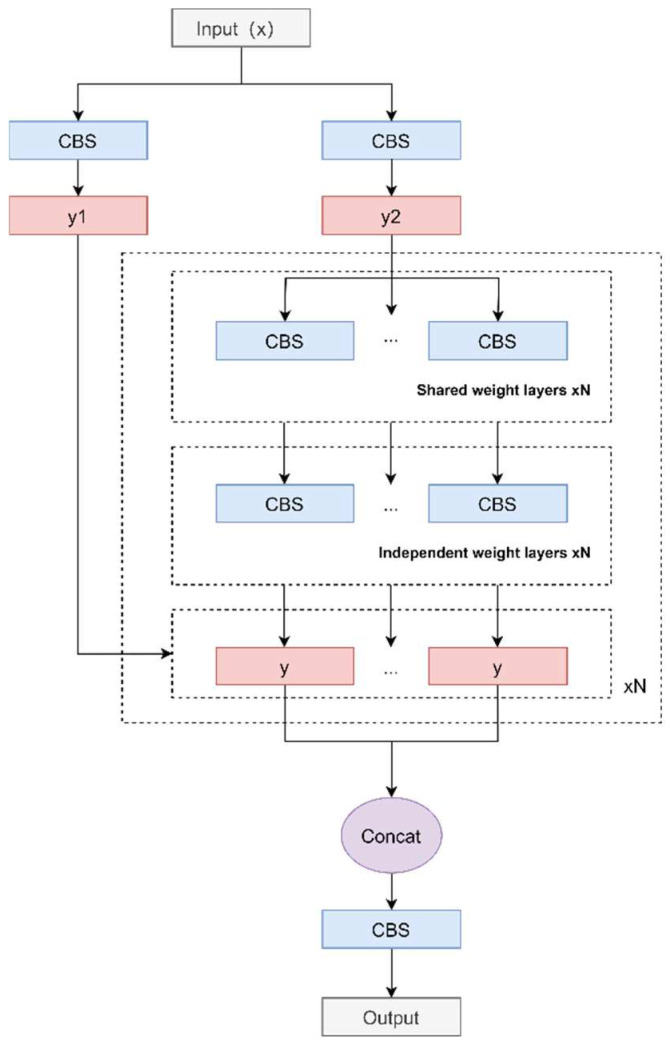
Structure of the RCSP.

**Figure 6 sensors-24-04491-f006:**
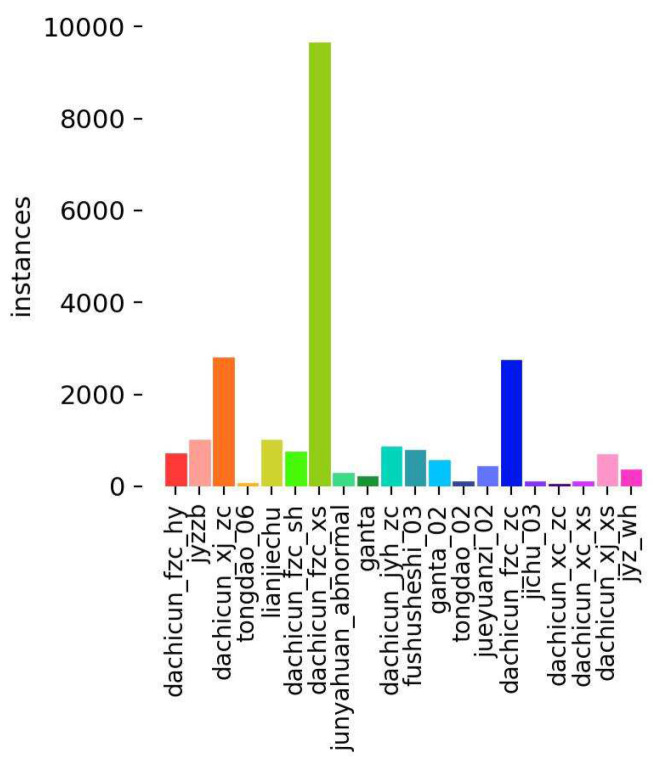
Distribution of the number of defective samples in the training set.

**Figure 7 sensors-24-04491-f007:**
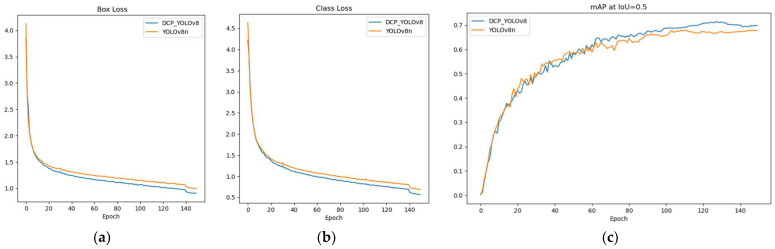
Analysis during training; (**a**) represents the box loss curve, (**b**) represents the classification loss curve, and (**c**) represents the map curve changes of YOLOv8n and DCP-YOLOv8.

**Figure 8 sensors-24-04491-f008:**
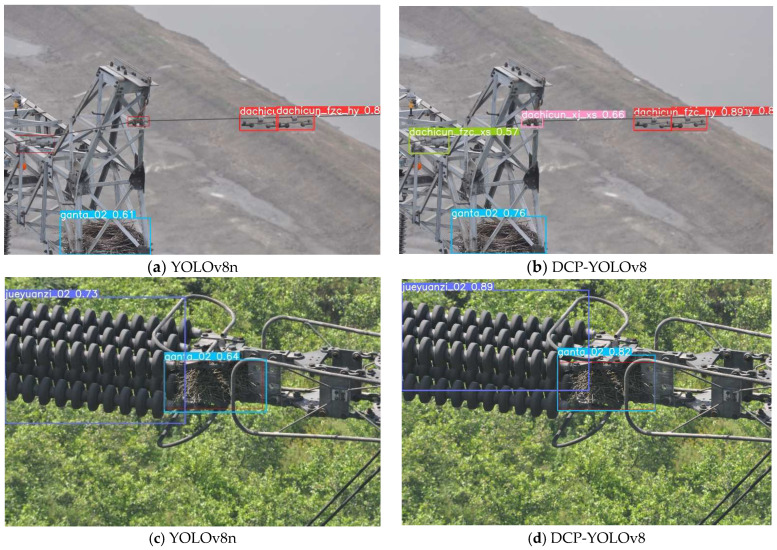
Some examples of defect detection effects.

**Figure 9 sensors-24-04491-f009:**
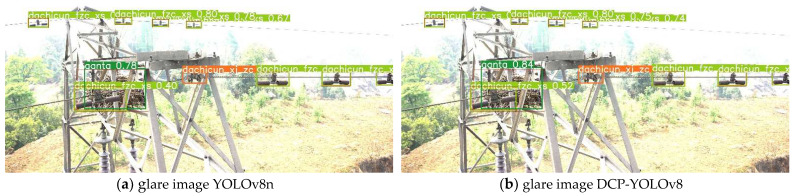
Robustness analysis of DCP-YOLOv8 in complex environments.

**Figure 10 sensors-24-04491-f010:**
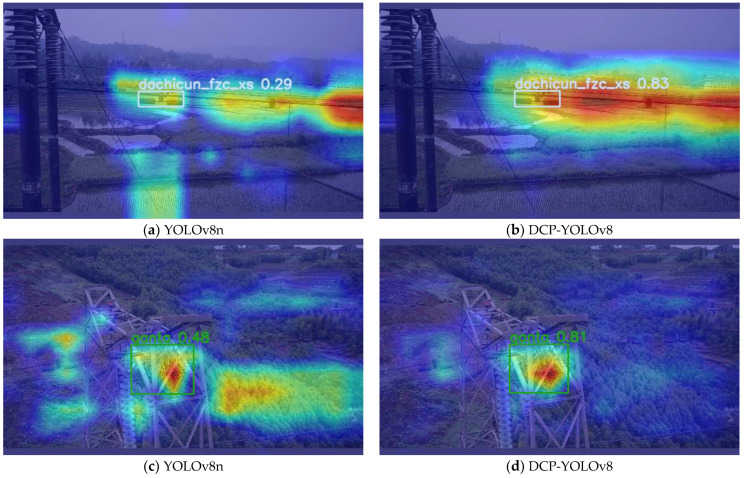
Some examples of defective target heat maps.

**Table 1 sensors-24-04491-t001:** Experiment environment parameters.

Experimental Conditions	Parameter
CPU model	Intel Xeon Silver 4208
GPU model	GeForce RTX 2080 Ti
Memory	22 G
operating system	Ubuntu 20.04
GPUaccelerator	CUDA 11.2
Driver version	460.80

**Table 2 sensors-24-04491-t002:** Some key parameters set during model training.

Train Parameter	Numerical Value
Epochs	150
Batchsize	16
Data augmentation strategy	Mosaic
Enter image size	640 × 640
Momentum	0.937
Weight_decay	0.0005

**Table 3 sensors-24-04491-t003:** Comparison of the effects of C2f_DCNv3 at different positions in YOLOv8.

C2f_DCNv3	mAP50/%	GFLOPs	Parameters	F1-Score/%
YOLOv8n	67.9	8.1	**3.1 M**	65.4
YOLOv8n + backbone	**69.4**	7.6	3.3 M	**67.5**
YOLOv8n + neck	68.7	**7.2**	3.2 M	66.8
YOLOv8n + backbone + neck	67.6	7.8	3.9 M	65.3

**Table 4 sensors-24-04491-t004:** Comparison experiments with different feature fusion modules.

Pyramid	mAP50/%	GFLOPs	Parameters	F1-Score/%
YOLOv8n	67.9	8.1	3.1 M	65.4
YOLOv8n + CSPNet	68.3	8.3	3.5 M	66.1
YOLOv8n + RCSP	**69.8**	**7.4**	**2.4 M**	**67.9**

**Table 5 sensors-24-04491-t005:** Comparison experiments of different detection head modules.

Detection Head	mAP50/%	GFLOPs	Parameters	F1-Score/%
YOLOv8n	67.9	**8.1**	**3.1 M**	65.4
YOLOv8n + Dyhead	68.7	21.1	5.7 M	67.1
YOLOv8n + DCNv3_Dyhead	**70.8**	17.8	5.1 M	**68.7**

**Table 6 sensors-24-04491-t006:** Ablation experiments.

Model	C2f_DCNv3	RCSP	DCNv3_Dyhead	mAP50/%	GFLOPs	Parameters
YOLOv8n				67.9	8.1	3.1 M
	√			69.4	7.6	3.3 M
		√		69.8	**7.4**	**2.4 M**
			√	70.8	17.8	5.1 M
	√	√		69.3	7.9	2.9 M
	√		√	70.2	10.3	4.1 M
		√	√	68.9	9.2	3.0 M
	√	√	√	**72.2**	8.9	2.8 M

**Table 7 sensors-24-04491-t007:** Performance comparison with mainstream detection algorithms.

Model	mAP50/%	GFLOPs	Parameters	F1-Score/%	FPS
Faster R-CNN [13]	64.8	179	60.1 M	61.2	11.9
RetinaNet [20]	65.4	173.8	54.9 M	62.3	12.5
RT-DETR [31]	67.7	60.2	20.1 M	64.8	40.3
Damo YOLO [32]	67.3	18.2	8.5 M	65.1	58.4
YOLOv6 [33]	66.8	11.4	4.7 M	63.4	72.3
YOLOv7 [34]	65.6	12.8	5.9 M	62.5	63.6
YOLOv8	67.9	**8.1**	3.1 M	65.4	**109.3**
DCP-YOLOv8	**72.2**	8.9	**2.8 M**	**69.5**	103.1

## Data Availability

Data are contained within the article.

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
