# Peer review of "A Lightweight and Efficient Multi-Type Defect Detection Method for Transmission Lines Based on DCP-YOLOv8"

_sensors, 2024, doi:10.3390/s24144491_

Round 1
Reviewer 1 Report
Comments and Suggestions for Authors
This paper introduces a lightweight and efficient multi-type defect detection method for transmission lines based on DCP-YOLOv8, aiming to enhance the accuracy and efficiency of defect detection on power transmission lines. The method effectively balances detection accuracy and performance with quantitative generalizability in the scenario of multi-type defect detection.
Strengths:
- Motivation: The research is motivated by a clear and pressing need for real-time and accurate detection in power transmission line maintenance, which is well-aligned with current industry demands.
- Novelty: The introduction of the C2f_DCNv3 module and RCSP structure represents innovative network design, enhancing the model's ability to recognize defects of varying scales and in complex backgrounds. The introduction of the C2f_DCNv3 module and RCSP structure represents innovative network design, enhancing the model's ability to recognize defects of varying scales and in complex backgrounds.
- Technical Correctness: The technical approach is based on cutting-edge deep learning theories and practices, such as the optimized version of YOLOv8, ensuring the state-of-the-art and correctness of the technical solution.
- Experimental Validation: Comparative experiments with other mainstream object detection algorithms establish the superiority of DCP-YOLOv8 in terms of accuracy, computational complexity, and parameter count.
- References: A substantial number of related studies are cited, providing solid support for the theory and experiments of the paper.
Weaknesses:
- Motivation: While the research motivation is clear, further discussions on the applicability and scalability of the method under different regional and environmental conditions may be necessary.
- Novelty: Although the method is novel, further exploration and validation may be needed regarding the diversity and complexity of detection in power transmission line maintenance.
- Technical Correctness: While the technical solution is based on advanced theories, more details may be needed to prove its portability and stability across different hardware platforms.
- Experimental Validation: Although the experiments validate the effectiveness of the method on a specific dataset, further testing in more diverse datasets and real-world scenarios is needed to prove robustness.
- References: While the literature review is comprehensive, there are some related works are missing. Particularly, some related work should be cited and compared with the proposed method, including: [1] DPNet: dual-path network for real-time object detection with lightweight attention. IEEE Transactions on Neural Networks and Learning Systems, early access, DOI: 10.1109/TNNLS.2024.3376563,2024. [2] Multiscale Shared Learning for Fault Diagnosis of Rotating Machinery in Transportation Infrastructures. IEEE Transactions on Industrial Informatics, vol. 19, no. 1, pp. 447-458, Jan. 2023, doi: 10.1109/TII.2022.3148289.
This paper introduces a lightweight and efficient multi-type defect detection method for transmission lines based on DCP-YOLOv8, aiming to enhance the accuracy and efficiency of defect detection on power transmission lines. The method effectively balances detection accuracy and performance with quantitative generalizability in the scenario of multi-type defect detection.
Strengths:
- Motivation: The research is motivated by a clear and pressing need for real-time and accurate detection in power transmission line maintenance, which is well-aligned with current industry demands.
- Novelty: The introduction of the C2f_DCNv3 module and RCSP structure represents innovative network design, enhancing the model's ability to recognize defects of varying scales and in complex backgrounds. The introduction of the C2f_DCNv3 module and RCSP structure represents innovative network design, enhancing the model's ability to recognize defects of varying scales and in complex backgrounds.
- Technical Correctness: The technical approach is based on cutting-edge deep learning theories and practices, such as the optimized version of YOLOv8, ensuring the state-of-the-art and correctness of the technical solution.
- Experimental Validation: Comparative experiments with other mainstream object detection algorithms establish the superiority of DCP-YOLOv8 in terms of accuracy, computational complexity, and parameter count.
- References: A substantial number of related studies are cited, providing solid support for the theory and experiments of the paper.
Weaknesses:
- Motivation: While the research motivation is clear, further discussions on the applicability and scalability of the method under different regional and environmental conditions may be necessary.
- Novelty: Although the method is novel, further exploration and validation may be needed regarding the diversity and complexity of detection in power transmission line maintenance.
- Technical Correctness: While the technical solution is based on advanced theories, more details may be needed to prove its portability and stability across different hardware platforms.
- Experimental Validation: Although the experiments validate the effectiveness of the method on a specific dataset, further testing in more diverse datasets and real-world scenarios is needed to prove robustness.
- References: While the literature review is comprehensive, there are some related works are missing. Particularly, some related work should be cited and compared with the proposed method, including: [1] DPNet: dual-path network for real-time object detection with lightweight attention. IEEE Transactions on Neural Networks and Learning Systems, early access, DOI: 10.1109/TNNLS.2024.3376563,2024. [2] Multiscale Shared Learning for Fault Diagnosis of Rotating Machinery in Transportation Infrastructures. IEEE Transactions on Industrial Informatics, vol. 19, no. 1, pp. 447-458, Jan. 2023, doi: 10.1109/TII.2022.3148289.
Author Response
很抱歉,此附件已提交给审稿人 2,并且不小心在此处提交。

Reviewer 2 Report
Comments and Suggestions for Authors
This study proposed a lightweight and efficient multi-type defect detection method for transmission lines based on DCP-YOLOv8. The research experiment is well-designed and useful for circuit defect detection. However, certain problems remain.
Major comment:
(1). Explain why a lightweight model can identify only a few defects?
(2). The article does not clearly describe what causes baseline's poor feature extraction.
(3). Why is the data divided according to 8:1.5:0.5?
(4). The introduction does not adequately describe the problem to be solved by the research. Most of the space is devoted to lightweighting, but DCP-YOLOv8 has limited parameter reduction compared to baseline.
Minor comment:
(1). Whether the results of the experiment were obtained in the test set or the validation set.
(2). Line 50-74, the logic is not clear, the core idea of the whole paragraph should be in the front.
(3). line365-371, confusing to read.
(4). Please bold the optimal values in the table.
(5). Figure 6 needs to be reformatted.
(6). Suggest adding GT to Figure 7.
Author Response
Thank you very much for your valuable comments. Please see the attachment.

Reviewer 3 Report
Comments and Suggestions for Authors
I recommend your article for publication with minimal but thorough technical correction.
My main comments relate mainly to the technical design of the text. “Sensors” magazine has a very simple and very convenient universal template for designing articles https://www.mdpi.com/files/word-templates/sensors-template.dot
Notes:
1. I would additionally include in the list of keywords: “machine vision”, “visualization”, “visible radiation”, “infrared radiation”.
2. Line 75, missing space before “[8]“.
3. Line 91, missing space before “[10, 11]“.
4. After captions to drawings it is customary to put “.”
5. Line 286, missing spaces after “[30]”, before “[31]“.
6. Formula (3) increased.
7. There are no punctuation marks after the formulas.
8. After the names of tables it is customary to put “.”
9. Check the figure numbering after Figure 6 and the corresponding figure references.
10. Replace the “x” sign in the sizes of rectangular and square objects with the “´” symbol (Table 2).
11. Spaces are required before references in Table 7.
12. In the magazine “Sensors” it is not customary to increase the spacing between paragraphs and the text across the width of the page.
13. Before captions for fragments of drawings, a space is required, for example, “(a) YOLOv8n”.
14. The formatting of links should be uniform. Design of links in MDPI style. The link to https://doi.org/... makes it easier for the curious reader to access the cited article.Wang, S.; Xia, X.; Ye, L.; Yang, B. Automatic Detection and Classification of Steel Surface Defect Using Deep Convolutional Neural Networks. Metals 2021, 11, 388. https://doi.org/10.3390/met11030388
Be careful when correcting the text of the article, maybe I missed something else.

Author Response

(The authors gave the same response as above.)

Round 2
Reviewer 2 Report
Comments and Suggestions for Authors
The authors have substantially improved the quality of the manuscript, and I recommend acceptance with minor revisions.
I have no suggestions other than the 8:1.5:0.5 data division issue.
I don't need the authors to give me science on what each set does after division. The point I am trying to make is that 8:1.5:0.5 is not a common division ratio, and I would like to ask why the authors did not use the common division ratios of 6:2:2 or 7:2:1, but instead chose such a unique ratio.
Author Response
thanks for your kind suggestion, please see the attachment
